# Ex Vivo Model of Neuroblastoma Plasticity

**DOI:** 10.3390/cancers15041274

**Published:** 2023-02-17

**Authors:** Paula Schäfer, Stefanie Muhs, Lucas Turnbull, Palwasha Garwal, Hanna Maar, Timur A. Yorgan, Eva Tolosa, Tobias Lange, Sabine Windhorst

**Affiliations:** 1Department of Biochemistry and Signal Transduction, University Medical Center Hamburg-Eppendorf, Martinistrasse 52, D-20246 Hamburg, Germany; 2Institute of Anatomy and Experimental Morphology, University Medical Center Hamburg-Eppendorf, Martinistrasse 52, D-20246 Hamburg, Germany; 3Department of Osteology and Biomechanics, University Medical Center Hamburg-Eppendorf, Martinistrasse 52, D-20246 Hamburg, Germany; 4Department of Immunology, University Medical Center Hamburg-Eppendorf, Martinistrasse 52, D-20246 Hamburg, Germany; 5Institute of Anatomy I, Jena University Hospital, Teichgraben 7, D-07743 Jena, Germany

**Keywords:** neuroblastoma, spontaneous metastases, xenograft model, tumor plasticity, immune response, cancer antigen, MAGE-A3

## Abstract

**Simple Summary:**

The complexity of tumor cell plasticity is still poorly understood. In particular, cellular changes during the metastatic process are difficult to monitor. This is a descriptive study of cell lines derived from primary tumors of xenografted LAN-1 cells and the corresponding three generations of bone metastases. Our results of ex vivo analysis of the cell lines depict the ability of tumor cells to adapt and survive in different microenvironments undergoing significant cellular alterations. The cell lines show strong phenotypical and biochemical changes and even an altered response to immune cells and chemotherapy. In conclusion, this mouse model allows to analyze the complex changes in tumor cell populations during metastasis and can be adapted to cell lines from different tumor origins.

**Abstract:**

Tumor plasticity is essential for adaptation to changing environmental conditions, in particular during the process of metastasis. In this study, we compared morphological and biochemical differences between LAN-1 neuroblastoma (NB) cells recovered from a subcutaneous xenograft primary tumor (PT) and the corresponding three generations of bone metastasis (BM I–III). Moreover, growth behavior, as well as the response to chemotherapy and immune cells were assessed. For this purpose, F-actin was stained, mRNA and protein expression assessed, and lactate secretion analyzed. Further, we measured adhesion to collagen I, the growth rate of spheroids in the presence and absence of vincristine, and the production of IL-6 by peripheral blood mononuclear cells (PBMCs) co-incubated with PT or BM I–III. Analysis of PT and the three BM generations revealed that their growth rate decreased from PT to BM III, and accordingly, PT cells reacted most sensitively to vincristine. In addition, morphology, adaption to hypoxic conditions, as well as transcriptomes showed strong differences between the cell lines. Moreover, BM I and BM II cells exhibited a significantly different ability to stimulate human immune cells compared to PT and BM III cells. Interestingly, the differences in immune cell stimulation corresponded to the expression level of the cancer-testis antigen MAGE-A3. In conclusion, our ex vivo model allows to analyze the adaption of tumor populations to different microenvironments and clearly demonstrates the strong alteration of tumor cell populations during this process.

## 1. Introduction

Neuroblastoma, the most common solid cancer in infants, derives from sympathetic nervous tissue and ranges from self-limiting to high-risk disease. High-risk disease patients have a 5-year survival rate of less than 50% and characteristically include metastasis to the bone [1]. Only certain cells within the primary tumor cell population are able to self-renew and form distant metastases. These cells are referred to as metastasis-initiating cells (MICs) and are proposed to have stem cell-like characteristics. Cancer stem cells are characterized by remarkable cellular plasticity and, depending on the microenvironment, are able to switch from a differentiated to an undifferentiated phenotype, and the other way around, making it difficult to treat them [2].

Neuroblastoma cell populations are highly plastic, and the tumors consist of different cellular subtypes. Initially, three main populations have been defined: the less tumorigenic neuroblastic cells (N-type), the non-tumorigenic substrate-adherent (S-type) cells, and the intermediate-type (I-like) cells, which are considered to have stem-cell properties. I-like type cells are 4 to 5 times more tumorigenic than N-type cells. N-type cells exhibit a neuron-like morphology, express neuronal marker proteins, and grow as floating spheroids. S-type cells grow as monolayers and have a non-neuronal phenotype. They are derived from glial cells. The I-like type cells express marker proteins of the N- and S-type and can switch to the N- or S-like status [3,4]. Recent research has furthered the model of I-/N- and S-type cells and revealed two main subtypes of neuroblastoma cells. One subtype of cells is adrenergic, neuron-like, and more differentiated, and the other exhibits a mesenchymal phenotype [5,6,7].

Since NB cell populations are heterogeneous, they mostly consist of N-, S-, and I-like types, and can be selected depending on the environmental conditions. This plasticity can strongly impede targeted tumor therapy and immune cell therapy, as the respective targets can be transiently up- and down-regulated or certain tumor populations selected. This depends on the current tumor state and on the interaction with the microenvironment [8].

Here, an ex vivo mouse model was applied to gain further insights into NB plasticity in relation to chemotherapy response and immune cell stimulation. For this purpose, cell lines were established in a former project [9]. The human NB cell line LAN-1 was transplanted subcutaneously (s.c.) onto immunodeficient *Rag2*^−/−^ mice to form xenograft primary tumors (PT) and spontaneous bone marrow metastases (BM I) [9]. PT and BM I were re-cultured, and BM I was s.c. re-injected to form BM II. BM II cells were once again s.c. re-injected and the corresponding bone marrow metastases (BM III) were re-cultured (see illustration in Figure 1 and [9]). With this cycling exposure to skin and bone marrow, we aimed to mimic the ability of NB cell populations to adapt to different environments.

## 2. Materials and Methods

### 2.1. Cell Culture and Preparation of Cell Lysates

In the former project, LAN-1-derived cell lines were derived from s.c. xenograft primary tumor as well as from spontaneous bone marrow metastasis generation, see Figure 1 and for a detailed description Labitzky et al. 2020 [6]. These cell lines were cultured in RPMI Medium 1640 (1×) + GlutaMAX^TM^ (Gibco #72400-021, Carlsbad, CA, USA) with 5% FCS (Gibco #10270106) and 100 U/mL penicillin and 100 µg/mL streptomycin (Gibco #15070063). Because of their dynamic characteristics, all four sublines were cultured for a maximum of six weeks after thawing to ensure reliable results. Mycoplasma tests were performed monthly, using the MycoAlert Mycoplasma Detection Kit from Lonza, Basel, Switzerland.

### 2.2. Western Blotting

To prepare cell lysates, cell suspensions were collected after splitting and centrifuged at 500× *g* for 5 min, resuspended with phosphate-buffered saline (PBS) for washing, and centrifuged again. The cell pellet was then dissolved in 200 µL M-PER lysis buffer (ThermoFisher # 78501, Waltham, MA, USA) and frozen at −20 °C overnight to disrupt the cells. To remove cell debris, the lysates were thawed on ice, centrifuged at 15,000× *g* at 4 °C for 20 min, and the supernatant was collected as the finished cell lysate. Cell lysates were stored at −20 °C. For Western blotting, the protein concentration of the lysates was determined using the Bradford assay (Bradford, 1976) with Protein Assay Dye Reagent (Bio-Rad #5000006, Hercules, CA, USA) and a BSA standard (ThermoFisher #23209). Samples were prepared with a final protein concentration of 1 µg/µL in SDS sample buffer (Laemmli buffer). These samples were heated to 95 °C for 5 min (Thermomixer comfort) and afterwards stored at −20 °C.

Western blotting was performed by standard procedures, using nitrocellulose membranes (ThermoFisher #88018; 3–4 h 45 V). Antibodies and conditions used are listed in Appendix A. For quantification, blots were performed threefold using three different lysates, imaged using chemiluminescence imagers (Intas ECL Chemocam, Vilber Fusion FX), and expression was analyzed using ImageJ. Significant changes were calculated using Student’s *t*-test.

### 2.3. F-Actin Staining by Phalloidin and F-Tractin

F-actin was stained with Alexa-fluor488-coupled phalloidin (dilution 1:1000 in PBS) for 1 h at room temperature in cells fixed with 4% paraformaldehyde. As an alternative method to label F-actin in living cells, LAN-1 cells were transfected with a vector coding for the actin-binding domain of ITPKA (F-tractin) [10]. For DNA transfection, 2.5 × 10^4^ cells were seeded onto chamber slides (Ibidi #80821, Gräfelfing, Germany) that were pre-coated with poly-L-lysine (Sigma P4707, Saint Louis, MO, USA) and grown to about 80% confluence. Then, DNA transfection was performed according to the K2 Transfection System protocol (Biontex T060, München, Germany). The reagents used were K2 Multiplier, K2 Transfection Reagent, Optimem Medium (Gibco 31985-062), and 300 ng DNA (GFP-marked F-tractin) per well. The transfection medium was changed to fresh medium 24 h before imaging.

### 2.4. BrdU Assay

To measure newly synthesized DNA (thus cellular division), cells are incubated with Bromdesoxyuridin (BrdU) that can be detected by antibody reaction. For this purpose, 1 × 10^4^ cells/well were seeded to two 96-well plates. After incubation for 24 h, 10 µM BrdU was added, and after further incubation for 24 or 48 h, the cells were dried, and BrdU incorporation was analyzed by antibody detection, according to the instructions of the supplier (Roche #11647229001, Rotkreuz, Switzerland). To calculate the proliferation rate, the values obtained for cells incubated with BrdU for 48 h were divided by those obtained after incubation for 24 h.

### 2.5. Measurement of Spheroid Growth and Migration

Five hundred microliters of 2% agarose was pipetted into micro-molds from Merck, Darmstadt, Germany (MicroTissues^®^ 3D Petri Dish^®^ micro-mold spheroid kit #Z764094), and after the agarose was gelled, it was removed and transferred to a dish on a 12-well plate. Thereafter, 2.5 mL of cell culture medium per well was added and incubated for 15 min; this step was repeated once. Then, 190 µL of a suspension containing 500,000 cells/mL was carefully applied drop-wise to the chamber. After incubation for 10 min, the wells outside of the chamber were filled with 2.5 mL of medium. Spheroids were formed after 24 h and imaged every 24 h for three days. Spheroid areas were measured by ImageJ.

To measure migration, the spheroids were grown in hanging drops for 5 days, transferred to 24-well plates, and covered with collagen I. Thereafter, the cells adhere to the bottom of the well and start to migrate. Migration was imaged by life cell imaging for 24 h, and the area of the migration zone was analyzed by ImageJ.

### 2.6. qPCR

For quantitative PCR analysis, cells were seeded on 10 cm dishes and grown to about 80% confluence. Cells were then harvested and washed once in PBS, centrifuging at 500× *g* for 5 min. RNA was isolated using the NucleoSpin^®^ RNA Kit (Macherey-Nagel #740955, Düren, Germany). Reverse transcription was performed using SuperScript^TM^ IV VILO^TM^ Master Mix with ezDNase™ (ThermoFisher #11766050) and 2 µg of the previously isolated RNA. For qPCR, a mastermix was prepared containing 5 µL of PowerUp™ SYBR™ Green Master Mix (ThermoFisher #A25742), 1.5 µL of each primer (3 µM; all primers from Integrated DNA Technologies, Coralville, IA, USA), and 1 µL of DEPC-treated water (Qiagen #129115, Venlo, The Netherlands) for each reaction. For quantification, 1 µL of cDNA (1 µL of DEPC-treated water for control) was analyzed with 9 µL of qPCR mastermix. QuantStudio 3 cycler (Applied Biosystems, Waltham, MA, USA) and QuantStudio^TM^ Design and Analysis Software were used. qPCR experiments were performed at least twofold.

### 2.7. Lactate Assay

Extracellular lactate concentrations were determined using the Lactate-Glo^TM^ Assay (Promega #J5021, Madison, WI, USA). Ten thousand cells were seeded onto a 96-well plate in 100 µL medium with dialyzed FCS (Gibco #11520646) to minimize lactate in the medium. Samples were collected 8, 24, 48, and 72 h after seeding and stored at −20 °C.

Medium samples were diluted 1:100, and the lactate concentrations were evaluated following Promega protocol, using Tecan plate reader infinite 200 pro for measuring luminescence.

### 2.8. Transcriptome Analysis

Microarray-based transcriptome analysis was performed using isolated RNA as described above (see Section 2.6). The concentration and integrity of RNA were determined using the NanoDrop ND-1000 system (NanoDrop Technology, Waltham, MA, USA) and the TapeStation 2200 system (Agilent Technologies, Santa Clara, CA, USA). Genome-wide expression analysis was performed utilizing the Clariom D assay in mice (Thermo Fisher Scientific, Waltham, MA, USA). In brief, 100 ng of total RNA per sample was used for the synthesis of proprietary labeled 2nd-cycle ss-cDNA. Further processing was performed according to the manufacturer’s GeneChip™ WT PLUS reagent kit manual (document 703174, revision A.0). Then, 5.5 µg of fragmented and labeled cDNA was used for microarray hybridization by incubation at 45 °C for 16 h. After washing the microarrays with an Affymetrix Fluidics Station 450, they were scanned with an Affymetrix GeneChip Scanner 7G. Data analysis was performed in the Transcriptome Analysis Console v. 4.0.1.36 (Thermo Fisher Scientific, Inc.) using default settings (version 1) and Gene + Exon − SST-RMA as summarization. Average fold-change values were calculated using Tukey’s bi-weight average algorithm.

### 2.9. IL-6 Assay

For the IL-6 assay, 3 × 10^5^ cells of each LAN-1-derived cell line were seeded in 250 µL medium onto 24-well plates, including medium-only control wells. Cells were incubated for 24 h at 37 °C to adhere. Peripheral blood mononuclear cells (PBMCs) were isolated from venous blood samples by gradient centrifugation (Lymphoprep^TM^, StemCell Technologies #07801, Vancouver, BC, Canada). PBMCs were seeded in RPMI culture medium. One hundred microliters of medium containing 3 × 10^6^ PBMCs (10:1 to LAN-derived cells) was added to the wells, including a negative control containing only RPMI. Cells were co-incubated for 1, 2, and 3 h. The culture medium was removed and collected at each time point; 14 µL of a 25 × protease inhibitor cocktail was added, and samples were frozen at −20 °C. Samples were collected from three independent assays and stored for a maximum of six weeks. The collected samples were analyzed in an IL-6 ELISA (R&D Systems, Human IL-6 DuoSet ELISA #DY206 and DuoSet ELISA Ancillary Reagent Kit 2 #DY008) according to the manufacturer’s protocol.

Intracellular cytokine production of IL-6 was assessed by flow cytometry. PBMCs were incubated with LAN cell-derived supernatants for 17 h, the last three hours in the presence of Brefeldin A (10 µg/mL) to prevent the secretion of cytokines. PBMCs were subsequently harvested, washed, and incubated with antibodies against lineage markers (CD14, CD123, CD3, and HLA-DR) for the identification of the immune cell populations. Dead cells were identified with Alexa Fluor 750 carboxylic acid succinimidyl ester (Molecular Probes A20011) and excluded from the analysis. T cells were identified as CD3+ CD4− events, B cells as CD14− DR+, plasmacytoid dendritic cells (pDC) as HLA-DR+ CD123hi, and monocytes as CD14+. After cell surface staining, cells were permeabilized and fixed in Perm/Fix buffer (Invitrogen, Waltham, MA, USA) and subsequently incubated with anti-IL-1beta, anti-IL-6, and anti-IFN alpha antibodies. All antibodies were purchased at BioLegend, San Diego, CA, USA. After 30 min incubation, cells were washed and analyzed at the FACS Celesta (BD Biosciences, San Jose, CA, USA). FlowJo software v10 was used for data analysis.

### 2.10. Statistical Analysis

Student’s *t*-test was applied for comparison between two groups using SigmaPlot. Here, the data obtained for PT were compared pairwise with BM I, with BM II, or with BM III. To compare transcriptome data, average fold-change values were calculated using Tukey’s bi-weight average algorithm, and differences in the activation of signal transcription pathways were analyzed by the software tool Ingenuity Pathway Analysis (IPA) from Qiagen.

## 3. Results

### 3.1. Morphology of Metastasis-Derived LAN-1 Cells Changes after Serial Injection

To analyze the cellular plasticity of human NB cells, cell lines that had been previously established by Labitzky et al. [9] were analyzed. Here, LAN-1 cells were s.c. injected into the neck of SCID mice, and a cell line was established from the primary xenograft tumor (PT) as well as from BM-derived NB cells (BM I). In addition, the BM-derived cells were again injected subcutaneously. These cells grew subcutaneously and formed the second generation of bone metastasis (BM II). Moreover, these BM-derived cells were established in cell culture (BM II) and again injected subcutaneously. Finally, the corresponding BM cells were cultivated and named BM III. Thus, one cell line from the first primary tumor (PT I) and three cell lines from three generations of bone metastasis (BM I–III) were established (Figure 1).

First, potential morphological differences between these cell lines were analyzed. For this, the actin cytoskeleton was labeled by Alexa-fluor488-coupled phalloidin and cell shape was assessed. This analysis revealed that PT cells were rhombic, while BM I cells extended long dendrite-like cellular protrusions. The morphology of BM II and III cells appeared like a mixture of PT and BM I cells (Figure 2A). To validate and quantify this finding, the cells were transfected with the F-actin marker F-tractin [10], and analyzed by confocal microscopy. Here, different confocal sections were imaged, and 3D reconstructions were performed. These images allowed to separate the cells from each other, and the number of cells were counted extending protrusions > 50 µM, in a total of 100 cells (Figure 2B,C for 3D-images). This evaluation revealed that in the BM I population: 6-fold, in the BM II population: 3-fold, and in the BM III population: 2-fold more cells had extended protrusions > 50 µM compared to PT.

In conclusion, in the first bone metastasis (BM I), cells were enriched for extending long, dendrite-like protrusions, while in the second (BM II) and third (BM III) generations of bone metastasis, the number of cells extending dendrite-like cellular protrusions decreased.

### 3.2. BM-Derived LAN-1 Cells Show No Clear Differentiation to N- or S-like Cells

The most striking characteristic of BM I cells was the extension of long dendrite-like protrusions, indicating that a re-differentiated cell population had been enriched in the bone marrow. To analyze this assumption, the expression of the neuron-specific intermediate filament neurofilament light chain (NFL) was analyzed by real-time PCR (Figure 3A) and by Western blotting (Figure 3B). This evaluation revealed that NFL was up-regulated 10.7 ± 2.3-fold in BM I, 2.5 ± 0.2-fold in BM II, and 3.9 ± 0.5-fold in BM III cells on the mRNA level compared to PT (Figure 3A). On the protein level, it was 7.9 ± 1.9-fold in BM I, 3.3 ± 0.6-fold in BM II, and 3.8 ± 1.2-fold in BM III (Figure 3B). Furthermore, immunofluorescence analysis revealed that NFL was mainly located in cellular protrusions (Appendix A).

These results indicated that after the first formation of bone metastasis, neuron-like cells [3,4] having an adrenergic phenotype [5,6,7], were selected. To analyze this assumption, further markers of adrenergic (tyrosine hydroxylase (TH), dopamine-beta-hydroxylase (DBH), transcription factors PHOX2B and GATA3) and mesenchymal (fibronectin 1 (FN1), vimentin (VIM), the transcription factor PRRX1, and the cell surface protein CD44) transcripts were analyzed by qPCR (Figure 3C). However, although in BM I cells PHOX2B and GATA3 were slightly (2-fold) up-regulated, also the mesenchymal marker PRRPX1 and CD44 were very strongly (80-fold) or weakly (2-fold) up-regulated compared to PT. Likewise, in BM II and BM III cells, the mesenchymal marker PRRX1 showed a strong, and FN1 and CD44 a weaker, up-regulation, but the mRNA expression of VIM was not altered. Moreover, the mRNA level of TH was strongly up-regulated in BM III compared to PT. From these results, we conclude that in the BM-derived cell lines, no particular subtype was enriched, and no definitive classification of either the adrenergic or mesenchymal cell type can be determined.

Next, we analyzed the potential adaptation of the bone marrow-derived cells to the conditions of the bone marrow. In the bone marrow, the oxygen concentration can decrease up to 1.3%, and most BM-derived cells exhibit an anaerobic metabolism and grow in suspension [11]. Therefore, we analyzed cellular division under normoxia (20% oxygen) and hypoxia (2%) as shown in Figure 4A, examined lactate secretion as a readout for anaerobic metabolism (Figure 4B), and examined adherence to collagen I (Figure 4C). The results of these experiments revealed that the BM-derived cell lines showed a lower reduction in the proliferation rate under hypoxic conditions compared to PT. Moreover, BM I and BM II cells exhibited the highest levels of extracellular localization, and the lowest ability to adhere to collagen I compared to PT. Thus, BM I and BM II cells are best adapted to the conditions of the bone marrow.

In summary, our data reveal that the first generation of BM-derived cells (BM I) exhibited a neuron-like morphology, but none of the BM-derived cell lines showed a specific up-regulation of adrenergic or mesenchymal marker transcripts. On the other hand, BM I and BM II cells were best adapted to the conditions of the bone marrow.

### 3.3. BM-Derived Cells Exhibit a Lower Proliferation Rate and a Weaker Response to Chemotherapy

Next, we analyzed potential differences in spheroid growth and migration from spheroids between PT and BM-derived cell lines. We found that the spheroid growth rate was significantly reduced in the BM-derived cell lines compared to PT cells, and among these, BM III cells showed the lowest growth rate (Figure 5A). However, migration from spheroids was similar between the cell lines (Figure 5B).

In order to show if, due to the different growth rates of the LAN-1 cell lines, the response to chemotherapy may be different, the spheroids were treated with 50 nM vincristine for 24 h, and spheroid growth was assessed every 24 h. Indeed, we found that 48 h after treatment, the growth rate of PT cells was strongest and that of BM II and BM III weakest reduced by vincristine. However, since also in absence of vincristine the BM II and BM III spheroids were smaller compared to the PT and BM I spheroids (Figure 5A upper panel), the absolute size of vincristine-treated NB-spheroids was similar (Figure 5C upper panel). Furthermore, 72 h after treatment, in all cell lines, mainly necrotic cells and only a few spheroids were detectable. In conclusion, the proliferation rate of the LAN-1 cell lines decreased from PT to BM III, and the high growth rate of PT spheroids was most inhibited by vincristine treatment compared to the BM-derived cell lines.

### 3.4. The Transcriptome of BM Generations

To reveal whether the LAN-1 cell lines used different proliferation-related pathways, a transcriptome analysis was performed with subsequent pathway analysis (see Appendix A).

We found that the overall transcriptome showed strong differences between the cell lines (Figure 6) and revealed that among the 15 pathways analyzed, the PI3K-Akt and the VEGF pathways showed significant alterations between PT and the BM-derived cell lines (Appendix A). However, these pathways were not clearly down-regulated in the BM generations compared to PTs, thus do not explain the reduced proliferation rate of the BM-derived cell lines.

Next, we analyzed the transcripts whose mRNA levels are altered in all BM generations compared to PT, and selected the transcript showing the highest alteration in BM I–III compared to PT cells. This transcript turned out to be the cancer-testis antigen melanoma antigen A3 (MAGE-A3) [12]. In BM I cells, MAGE-A3 was up-regulated 31-fold, in BM II 61-fold, and in BM III 15-fold relative to PT (Figure 7A). Validation of these results by RT-PCR confirmed this pattern (Figure 7B).

In addition to its role as a tumor-specific antigen, MAGE-A3 exhibits oncogenic activity. It binds and stimulates the E3 ligase tripartite motif-containing protein 28 (TRIM28) [13], resulting in the degradation of the tumor suppressor genes p53 and AMPK [14]. Since p53 is not expressed in LAN-1 cells [15], the levels of AMPK and pAMPK were analyzed by Western blotting. As shown in Figure 8B, no differences were found between PT and BM I/BM II cells, while BM III cells had slightly down-regulated AMPK and pAMPK level. A possible explanation for this unexpected result was that, after the degradation of AMPK, its transcription was up-regulated. Indeed, our transcriptome analysis revealed a strong up-regulation of AMPK mRNA in BM I and BM II cells, showing the strongest MAGE-A3 signals (Figure 8B). Thus, cells with very high MAGE-A3 levels (BM I and BM II) seemed to compensate for MAGE-A3-mediated protein degradation by up-regulating its mRNA.

To sum up, the transcriptome of BM cell lines shows strong differences, and the transcript up-regulated the highest in all BM cell lines compared to PT is MAGE-A3. However, the cellular activity of MAGE-A3 seems to be compensated by the up-regulation of counter transcripts.

### 3.5. LAN-1 Cell Lines Show Strong Differences in Their Ability to Inhibit Immune Cell Stimulation

Our data, depicted in Figure 7 and Figure 8, show that up-regulation of MAGE-A3 in BM cell lines induced counter-regulation, indicating that MAGE-A3 is not a suitable target for targeted tumor therapy.

However, in addition to acting as a cellular master switch, expression of the cancer-testis antigen MAGE-A3 has been shown to strongly alter immune cell stimulation. Its presentation on the cell surface can result in the stimulation of T-cells [16], but it has also been shown that MAGE-A3 inhibits the secretion of cytokines and impedes the expression of MHC-I on the cell surface [17]. Based on these considerations, we analyzed the effect of NB cells on immune cell responses. First, we tested the cytokine production (IL-1β, IL-6, and IFNα) of peripheral blood mononuclear cells (PBMCs) co-incubated with cancer cells by FACS analysis. Among these, IL-6, which is produced by monocytes and dendritic cells, showed the highest level [18].

Based on this result, IL-6 secretion into the medium was analyzed after incubating PBMCs alone and in co-culture with NB cells. This analysis revealed that IL-6 secretion from PBMCs increased 3-fold from 2 h to 3 h of incubation (Figure 9A). Interestingly, this secretion significantly decreased in the presence of LAN-1 cells, among which BM cells with high MAGE-A3 expression (BM I and BM II) showed the strongest IL-6 suppression (Figure 9B).

In summary, these data show that LAN-1 cells inhibit the activity of IL-6-secreting immune cells, and this effect showed the same pattern as MAGE-A3 expression.

## 4. Discussion

Therapy success in the treatment of malignant cancer cells is difficult to predict because metastatic tumors are highly plastic in order to adapt to different environmental conditions. During progression from primary to metastatic disease, tumor cells can be completely reprogrammed or certain tumor cell populations selected, although the genomic mutations only show slight differences between primary tumors and metastases [19]. Therefore, it is believed that the different response of tumor cells to targeted therapies results from tumor plasticity [8]. Since it is not possible to analyze tumor plasticity in patients, in this study a mouse model was applied, allowing to assess biochemical and phenotypic changes during tumor progression. For this purpose, cell lines from xenotransplanted primary NB tumors as well as from three generations of bone marrow-derived cell lines (Figure 1 and [9]) were analyzed for phenotypic and biochemical changes.

Our key findings are that the spheroid growth rate of NB cell lines derived from the bone marrow decreased from the first (BM I) to the third (BM III) BM generation, and accordingly, initially, BM III cells showed the lowest response to treatment with the chemotherapeutic agent vincristine. This result is in line with the finding that tumor cells disseminating to the bone marrow are often slow-growing cells and therefore weakly respond to chemotherapy [20]. Future studies will show if this also holds true for other chemotherapeutics.

In addition, we showed that the first two generations of bone metastasis (BM I and BM II) are best adapted to the conditions of the bone marrow, while this adaptation diminished in BM III. However, although we first speculated that the high adaption of BM I cells to the conditions of the bone marrow is accompanied by the selection of adrenergic cells due to their neuron-like morphology and high NFL expression, a detailed analysis of marker transcripts did not reveal a clear differentiation of BM populations to adrenergic or mesenchymal cells [3,4,7]. Thus, we assume that the morphological and transcriptome differences between the NB cell lines reflect their transient adaption to the respective microenvironments. Future mechanistic studies will show if this assumption holds true.

Interestingly, the transcript with the highest up-regulation in the BM generations relative to PT (MAGE-A3) has also been shown to be up-regulated in bone marrow samples derived from non-small-cell lung cancer patients [21]. However, so far, it is not known if the expression of MAGE-A3 in cancer cells specifically increases in the bone marrow. Since in normal cells, MAGE-A3 transcription is blocked by methylation [12], it would be interesting to screen for factors in the bone marrow mediating the demethylation of the MAGE-A3 promoter.

The finding that MAGE-A3 interacts with the ubiquitin ligase TRIM28 and thereby stimulates proteasomal degradation of the tumor suppressors p53 and AMPK [12] makes MAGE-A3 an interesting target for tumor therapy. However, unexpectedly, analysis of the MAGE-A3 target AMPK showed no differences in AMPK protein concentration between PT and the BM-derived cell lines, and our transcriptome analysis revealed a strong AMPK mRNA up-regulation in cells with the highest MAGE-A3 expression (BM I and BM II). Thus, the cells seem to respond with counter-regulation, compensating for the cellular consequences of MAGE-A3 up-regulation. This finding is of high interest because it shows that up-regulation of master switch proteins such as MAGE-A3 can lead to counter-regulations that are difficult to predict. This principle has also been described for blockage of further cancer-related signal transduction pathways; i.e., the PI3K pathway (reviewed in [22]). Hence, targeting proteins with oncogenic potential does not always result in the desired effect, making it necessary to monitor the cellular consequences after therapeutic intervention.

In addition to its oncogenic activity, MAGE-A3 is a cancer-testis antigen, making it an apparently very suitable target for immune cell therapy. Therefore, we assumed that up-regulation of the protein may result in an increased immune cell response. However, the measurement of IL-6 secretion by PMBCs revealed the opposite. The BM cell lines with the highest MAGE-A3 expression (BM I and BM II) most strongly inhibited the IL-6 response. This result can be partly explained by the finding of Wang et al. showing that high MAGE-A3 expression in human melanoma cells inhibits chemokine secretion [17]. It is therefore very likely that also in vivo, MAGE-A3 suppresses the immune response by inhibiting the chemokine-mediated attraction of dendritic cells, which are essential for CD8 and CD4 T-cell differentiation [23]. This conclusion may also explain why two large phase III studies, validating the suitability of MAGE-A3 as an antigen, failed [24,25]. Future experiments with MAGE-A3 knockdown cells will elucidate the complex role of MAGE-A3 in stimulating or repressing the immune cell response.

## 5. Conclusions

In conclusion, the results obtained by our mouse model demonstrate the complex and often unexpected alterations of tumor populations during cycling exposure to different microenvironments. These findings emphasize the relevance of studying the complex cellular responses to different microenvironments, in particular in relation to cancer vaccines and therapeutics. Our mouse model provides the possibility to carefully monitor the reactions of tumor cells during metastasis as well as during therapeutic intervention by ex-vivo experiments. This model can be adapted to different tumor types and metastatic sites and may help improve the treatment of patients suffering from malignant tumors.

## Figures and Tables

**Figure 1 cancers-15-01274-f001:**
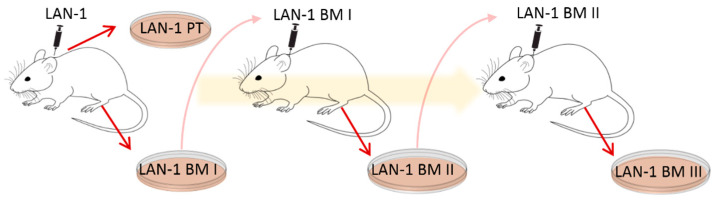
Scheme showing the serial injection of LAN-1 cells.

**Figure 2 cancers-15-01274-f002:**
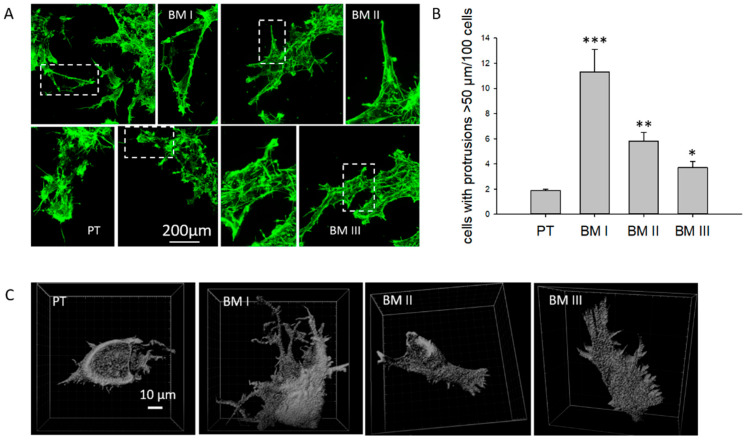
Morphological differences between PT and BM generations. (**A**) F-actin was labeled by Alexa-fluor488 coupled phalloidin (green), and the cells were analyzed by confocal microscopy. The figure includes an overview and magnified images of cell protrusions of each cell line (white frames). (**B**,**C**) Cells were transfected with F-tractin to label F-actin, imaged by confocal microscopy, and confocal sections were reconstructed to 3D images. (**B**) From these images, the number of cells having protrusions longer than 50 µm was counted. Shown are mean values + SD from 100 cells analyzed in four different experiments. * *p* < 0.05, ** *p* < 0.001, *** *p* < 0.0001.

**Figure 3 cancers-15-01274-f003:**
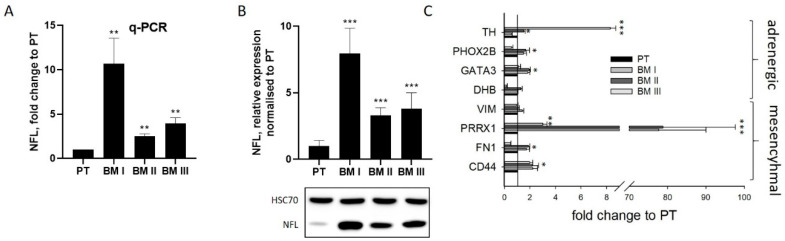
BM I cells show a differentiated phenotype. The levels of NFL transcripts and proteins were analyzed by real-time PCR (q-PCR, (**A**)) and by Western blotting (**B**). The uncropped blots are shown in Appendix A. (**C**) Analysis of adrenergic and mesenchymal marker transcripts by q-PCR. The levels obtained for PT were set to 1. Shown are mean ± SD of three independent experiments. * *p* < 0.05, ** *p* < 0.001, *** *p* < 0.0001.

**Figure 4 cancers-15-01274-f004:**
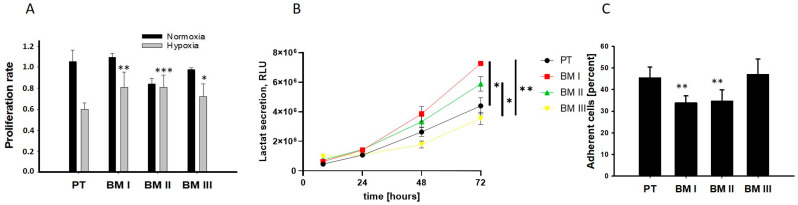
(**A**) The cells were cultivated under normoxia (20% O_2_), or hypoxia (2% O_2_), and 24 h after seeding, they were treated with BrdU. BrdU incorporation was assessed after further 24 h and 48 h of incubation. To calculate the proliferation rate, values obtained for cells incubated with BrdU for 48 h were normalized to those incubated for 24 h. (**B**) Lactate levels were analyzed in the medium of cells cultivated under normoxia after 12 h, 24 h, 48 h, and 72 h of incubation, using a luminescence-based assay. Luminescence is expressed as RLU. (**C**) The cells were seeded on collagen I-coated Petri dishes, and 4 h after incubation under normoxia, the number of adherent cells was counted. Shown are mean ± SEM of three independent experiments. * *p* < 0.05, ** *p* < 0.001, *** *p* < 0.0001.

**Figure 5 cancers-15-01274-f005:**
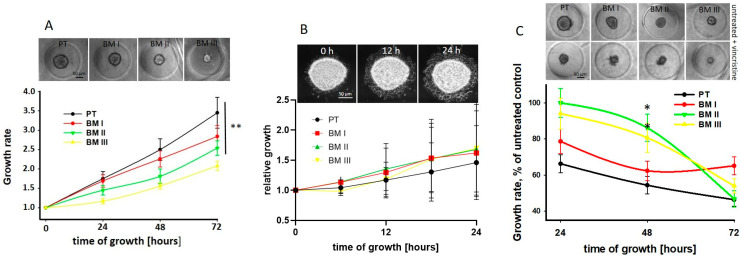
BM-derived cells show a lower proliferation rate and a weaker response to vincristine. (**A**) Spheroids were grown in agarose, and their growth rate was analyzed for 72 h, every 24 h, and the area of spheroids was determined by ImageJ. 16 spheroids were analyzed per experiment, and shown are mean ± SD of three independent experiments. ** *p* < 0.01. Images show representative spheroids 72 h after incubation. (**B**) Spheroids were grown in hanging drops, and after 5 days of incubation, spheroids were transferred to 24-well plates embedded with collagen I, and migration on the bottom of the wells was analyzed by live cell imaging using the Olympus Ixplore Live System. Four spheroids were analyzed per experiment, and shown are mean ± SD of three independent experiments. Shown are exemplary spheroids from PT cells, 0 h, 12 h, and 24 h after incubation. (**C**) Spheroids were treated with DMSO (vehicle control) or with 50 nM vincristine, and their areas were measured every 24 h for 72 h by ImageJ “growth rate”. The values obtained for control cells were set to 100%, and those for vincristine-treated cells were calculated. Images show representative untreated and vincristine-treated spheroids after 48 h of incubation. Scale bar: 50 µm. Shown are mean ± SEM of three independent experiments. * *p* < 0.05.

**Figure 6 cancers-15-01274-f006:**
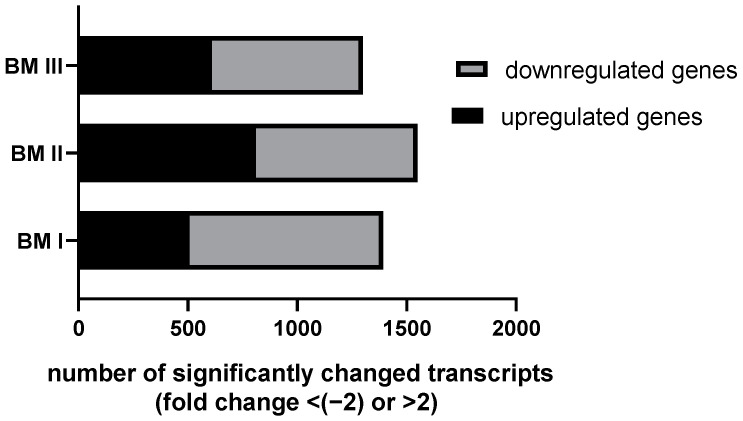
The transcriptome of PT and BM I–III was analyzed and up or downregulation of transcripts BM sublines compared to PT were calculated.

**Figure 7 cancers-15-01274-f007:**
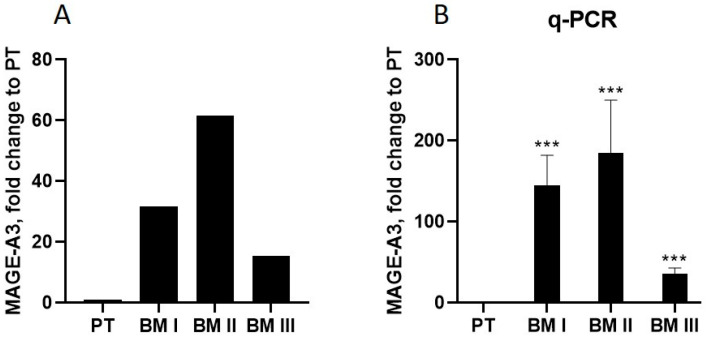
(**A**) The transcript showing the highest up-regulation in transcriptome analysis in the BM generations compared to PT was MAGE-A3. (**B**) Up-regulation of MAGE-A3 was confirmed by qPCR. Shown are mean + SEM of three independent experiments. *** *p* < 0.0001.

**Figure 8 cancers-15-01274-f008:**
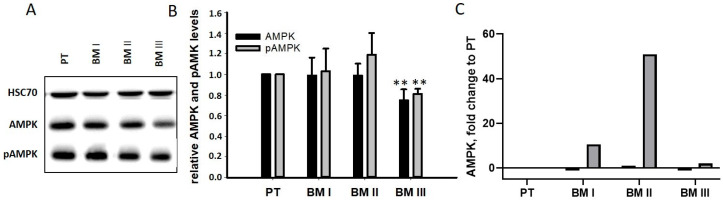
AMPK analysis. (**A**,**B**) Expression and phosphorylation (pAMPK) of AMPK, a target of MAGE-A3 [14], was analyzed by Western blotting. The uncropped blots are shown in Appendix A. (**B**) Shows densitometric analysis of the AMPK blots with mean + SD of three independent experiments. ** *p* < 0.005. The data obtained for PT were set to one. (**C**) Compensatory up-regulation of AMPK at the mRNA level was found in the transcriptome data.

**Figure 9 cancers-15-01274-f009:**
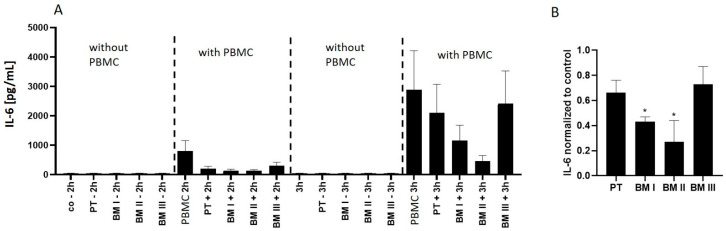
BM-derived cells inhibit immune cell activation. (**A**) Activation of PBMCs was analyzed by IL-6 response in the medium after co-incubation with NB cells (“with PBMC”) for 2 h or 3 h. PBMCs without NB cells were also analyzed as control (“PBMC”). In addition, tumor cells without PBMCs were assessed (“without PBMC”). (−): NB cells without PBMCs; (+) NB cells with PBMCs. (**B**) The IL-6 concentration after 3 h of incubation obtained from NB cells incubated with PBMCs was normalized (set to 1) to the IL-6 concentration measured in the samples containing PBMCs only. Shown are mean values ± SD of three independent experiments, including independent PMBC preparations, * *p* < 0.05.

## Data Availability

All data can be found in the text.

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
