# Peer review of "Ex Vivo Model of Neuroblastoma Plasticity"

_cancers, 2023, doi:10.3390/cancers15041274_

Round 1

Reviewer 1 Report (Previous Reviewer 1)

This manuscript has improved since the previous submission. Specifically, the analysis of additional marker genes in the PT and BM I-III clones reveals a more complex change in NB cell phenotypes as these tumor cells adapt to the bone marrow microenvironment. However, in some areas the writing and figure legends could be clarified and a number of instances remain where the conclusions are not fully supported by the data. Point by point critiques are listed, below:

On page 6, the authors indicate that “immunofluorescence analysis revealed that NFL was mainly located in cellular protrusions (Fig. S1)”, but Figure S1 appears to be un-cropped images of gels. Where is the supplemental data? Supplementary Table 1 with the transcriptome analysis also seems to be missing.

The legend for Fig. 3C should be revised to indicate that q-PCR is measuring mesenchymal transcripts or mRNAs, not proteins.

Following the description of the q-PCR data, the authors state that “These data indicate that in the BM-derived cell lines cells with renewable characteristics were enriched.” This conclusion is confusing since the previous analysis was focused on adrenergic and mesenchymal cell marker expression rather than assays of cell renewal or stemness. Similarly, on page 6, the authors conclude that “none of the BM-derived cell lines showed a specific up-regulation of adrenergic or mesenchymal marker proteins”, yet no data on mesenchymal and adrenergic proteins is included.

In the legend for Fig. 4 the authors state that "The cells were cultivated under normoxia or hypoxia (2% O2)" but only one set of data points is shown for each cell lines (PT, BMI-III). Were these cells grown in hypoxic or normoxic conditions? It would be easier to understand this data if cell counts, lactate levels, and other data points collected under hypoxic and normoxic conditions were shown separately.

The graph in Fig. 5C is difficult to understand. Is it possible to show the spheroid size (similar to Fig. 5A) +/- vincristine for each cell line?

Author Response

Dear Reviewer,
Thank you again for the suggestions to improve our manuscript “Ex vivo model of neuroblastoma cell plasticity”. We included your suggestions in the new version of the manuscript (marked in red).
Following please find my point-by-point reply.

Comment: On page 6, the authors indicate that “immunofluorescence analysis revealed that NFL was mainly located in cellular protrusions (Fig. S1)”, but Figure S1 appears to be un-cropped images of gels. Where is the supplemental data? Supplementary Table 1 with the transcriptome analysis also seems to be missing.

Answer: We apologize that we did not upload the supplementary data. This has been done now.

Comment: The legend for Fig. 3C should be revised to indicate that q-PCR is measuring mesenchymal transcripts or mRNAs, not proteins.

Answer: Thank you for noticing, we did this.

Comment: Following the description of the q-PCR data, the authors state that “These data indicate that in the BM-derived cell lines cells with renewable characteristics were enriched.” This conclusion is confusing since the previous analysis was focused on adrenergic and mesenchymal cell marker expression rather than assays of cell renewal or stemness. Similarly, on page 6, the authors conclude that “none of the BM-derived cell lines showed a specific up-regulation of adrenergic or mesenchymal marker proteins”, yet no data on mesenchymal and adrenergic proteins is included.

Answer: Yes, we agree, the stemness theory is overdone, thus was deleted this. Furthermore, we changed “proteins” to “transcripts”, please see page 8 and 12.

Comment: In the legend for Fig. 4 the authors state that "The cells were cultivated under normoxia or hypoxia (2% O2)" but only one set of data points is shown for each cell lines (PT, BMI-III). Were these cells grown in hypoxic or normoxic conditions? It would be easier to understand this data if cell counts, lactate levels, and other data points collected under hypoxic and normoxic conditions were shown separately.

Answer: Yes, we see, this has been done for BrdU measurements of cells grown under hypoxic or normoxic conditions (Figure 4a). Here, we wanted to measure if the proliferation rate under hypoxic conditions is different between PT and BM-cells. In case of lactate secretion and adhesion the cells were cultivated under normal conditions. Please see page 3, 7 and 8.

Comment: The graph in Fig. 5C is difficult to understand. Is it possible to show the spheroid size (similar to Fig. 5A) +/- vincristine for each cell line?

Answer: Yes, we did this and also changed the text accordingly because now one can nicely see that the absolute size of spheroids is similar between the cell lines.

Reviewer 2 Report (Previous Reviewer 2)

I am content with the changes made. 

Author Response

Dear Reviewer,
Thank you again for your review.

This manuscript is a resubmission of an earlier submission. The following is a list of the peer review reports and author responses from that submission.

Round 1

Reviewer 1 Report

In their study, Shafer et al. generated bone-marrow derived neuroblastoma cell lines by serial subcutaneous transplantation of LAN-1 cells followed by isolation of cells from the bone marrow (BM). They then go on to perform various cell phenotyping assays to characterize the BM-derived cell lines. These BM cell lines could be a potentially useful research model that would be of interest to cancer researchers with a focus on neuroblastoma and cancer progression. However, there are numerous areas where the data do not adequately support the authors’ conclusions. Additional experiments and analyses are necessary before this manuscript will be suitable for publication. Specific comments, below”

The data in Fig. 2 do not adequately support the conclusion that there is a morphological change in the BMI, II, and III cells. First, based on the images is it difficult to see how individual cells were distinguished from each other. Second, the quantification of projections does not appear to have been repeated and no statistics are reported. Third, the number of cells scored to determine the % of cells with longer protrusions was not indicated.

In Fig. 3 a single marker, NFL (NEFL?) is used as an indicator of a neuronal-like phenotype analyzed by RT-PCR and Western blotting. Are additional markers of neuroblastoma differentiation up-regulated as well? What about the status of stem and differentiation markers?

As presented, the data in Fig. 4 are confusing since the authors indicate that they compared normoxic and hypoxic growth conditions, yet these data are not distinguished on these graphs. Reporting the data for lactate secretion and BrdU incorporation for normoxia and hypoxia separately is more appropriate. As shown it is not clear whether the different LAN-1 cell lines show differential responses to hypoxia. In

Fig. 4A and Fig. 5A, what statistical test was used to determine whether the change in growth rate was significant, an ANOVA?

In Fig. 5A how many spheroids were scored per condition to determine the growth rate of the spheroids? Is it possible to disassociate these spheres and obtain a more accurate cell count. Finally, including representative photos of the PT versus BM cell.

In Fig. 6 it is not clear why more information about the differentially expressed genes is not included in the manuscript. Are there additional genes that are up- and down-regulated in the different cell lines beyond MAGE-A3? Analysis at the gene set level only is not sufficient. Do these data support the finding in Fig. 2 that NFL is differentially expressed?

In Fig. 8a, representative WB should be included to show the level of pAMPK phosphorylation. Unless a titration curve of protein input was used, it is not clear whether these analyses are in the linear range. Data in Fig. 8b do not seem to have been replicated as there are no error bars or statistics reported.

The graph in Fig. 9a should be labelled to indicate which conditions either included or excluded PBMCs.

In Fig. 10, why not just report the spheroid size or relative growth rate +/- vincristine separately? The normalization used here is confusing.

Reviewer 2 Report

The authors, Schäfer et al, have in this study established different cell lines as ex vivo neuroblastoma models. This was done by subcutaneous (sc) engraftment of LAN-1 neuroblastoma cells in mice, the authors used retrieved cells from both the primary sc tumor and the bone marrow to establish cell lines that were used again in mice similarly, in three generations. The cells were then characterized. The models can be used as tools in future studies to understand neuroblastoma plasticity.

1. The description of all animal studies is missing in the material & methods, also how the cell lines from the mice tumors were established and passaged. Thus, a lot of details are missing in regard to how the experiments were done, when did they harvest the tumors? At a specific size? How was the BM retrieval done? Etc etc. 

2. Did the authors confirm the cell identity of cells retrieved from the mice in regard that they were neuroblastoma cells (particularly the BM cells)? For example, by staining for PHOX2B.

2. The authors do not mention/discuss the adrenergic/mecenchymal tumor cell phenotype which has been and is a hot topic in the field of neuroblastoma (van Groningen et al, 2017, Boeva et al, 2017, Gautier M et al, 2021 etc). That is an important aspect of neuroblastoma plasticity, the transcriptome analysis can maybe be possible to use to characterize the cell phenotype of the different cell lines with the signatures provided in van Groningen et al, 2017 (doi: 10.1038/ng.3899). This would provide important insights. 

.  

3. The authors discuss differentiation but only show data one differentiation marker. Is it possible to get others from the transcriptome analysis and add to the manuscript? Otherwise, qPCR can be used to analyze this.

4. In the text the authors refer to different PT, why is only one included in all figures? Please clarify.

5. The authors state “PT cells reacted most sensitive to the chemotherapeutic agent vincristine. ” - it looks like in Fig 10 that after 3 days the effect of vincristine was the same for PT, BMI and BM III? Did the authors measure size or cell death in the spheroids as an additional measurement of chemotherapeutic effects? Or can this be done? Also, were any other chemotherapeutic drugs tested beside vincristine? 

6. If possible, please provide supplementary data in another format the “rar”, preferable pdf. 

Reviewer 3 Report

In this manuscript, Schäfer et al. applied an ex vivo model for analysis of human tumor plasticity in association with therapy response. For this reason, they recovered LAN-1 neuroblastoma (NB) cells from a subcutaneous xenograft primary tumor (PT) and three generations of bone metastasis (BM I - III). Their analysis of PT and the three BM generations revealed that their growth rate decreased from PT to BM III, and accordingly PT cells reacted most sensitive to the chemotherapeutic agent vincristine. In addition, they revealed that the differentiation status as well as the transcriptomes showed strong difference between the cell lines. Moreover, BM I and BM II cells were shown to exhibit a significantly different ability to stimulate human immune cells compared to PT and BM III cells, and they also showed that the differences in immune cell stimulation corresponded to the expression level of the cancer-testis antigen MAGE-A3. Their study shows phenotypical and biochemical changes in the cell lines and even altered suppression of immune cell response to the tumor cells.

This study allows us to better understand the elusive and complex nature of neuroblastoma cancer, as well as the highly complex tumor cellular environment. It has also been an effective study in showing the importance of monitoring cells in order to detect underlying cellular changes and make necessary adjustments in treatment in neuroblastoma cases that often respond poorly or not at all to chemotherapy. Experimental design is quite well done. They are very detailed and thoroughly designed to examine all aspects. The language of explanation is very simple and understandable. The flow of the narrative is very logical. Just like reading an exciting novel or story, there is an experimental setup and narration that is read with interest from start to finish. The discussion of the results is very rational. Noting the shortcomings of the study, however, they emphasized the importance of such an animal model in terms of monitoring the metastatic state in different cancers and the changes that cells undergo to adapt to their environment.

There is only one point that I would like to draw attention to: the generation of the ex vivo animal model and the generation of the LAN-1 PT and LAN-1 BMI-III cell lines may be better given in the material method section.